# S100B Serum Levels in Chronic Heart Failure Patients: A Multifaceted Biomarker Linking Cardiac and Cognitive Dysfunction

**DOI:** 10.3390/ijms25169094

**Published:** 2024-08-22

**Authors:** Jan Traub, Michael K. Schuhmann, Roxanne Sell, Stefan Frantz, Stefan Störk, Guido Stoll, Anna Frey

**Affiliations:** 1Department of Internal Medicine I, University Hospital Würzburg, 97080 Würzburg, Germany; frantz_s@ukw.de (S.F.); frey_a@ukw.de (A.F.); 2German Comprehensive Heart Failure Center, University Hospital Würzburg, 97087 Würzburg, Germany; 3Department of Neurology, University Hospital Würzburg, 97080 Würzburg, Germany; schuhmann_m@ukw.de (M.K.S.); stoll_g@ukw.de (G.S.); 4Department of Psychiatry, Psychosomatics and Psychotherapy, University Hospital Würzburg, 97080 Würzburg, Germany; sell_r@ukw.de

**Keywords:** S100B, heart failure, blood–brain barrier, cognition, mild cognitive impairment

## Abstract

S100 calcium-binding protein B (S100B) is a protein primarily known as a biomarker for central nervous system (CNS) injuries, reflecting blood–brain barrier (BBB) permeability and dysfunction. Recently, S100B has also been implicated in cardiovascular diseases, including heart failure (HF). Thus, we investigated serum levels of S100B in 146 chronic HF patients from the Cognition.Matters-HF study and their association with cardiac and cognitive dysfunction. The median S100B level was 33 pg/mL (IQR: 22–47 pg/mL). Higher S100B levels were linked to longer HF duration (*p* = 0.014) and increased left atrial volume index (*p* = 0.041), but also with a higher prevalence of mild cognitive impairment (*p* = 0.023) and lower visual/verbal memory scores (*p* = 0.006). In a multivariable model, NT-proBNP levels independently predicted S100B (T-value = 2.27, *p* = 0.026). S100B did not impact mortality (univariable HR (95% CI) 1.00 (0.99–1.01); *p* = 0.517; multivariable HR (95% CI) 1.01 (1.00–1.03); *p* = 0.142), likely due to its reflection of acute injury rather than long-term outcomes and the mild HF phenotype in our cohort. These findings underscore S100B’s value in comprehensive disease assessment, reflecting both cardiac dysfunction and potentially related BBB disruption.

## 1. Introduction

S100 calcium-binding protein B (S100B) is a member of the S100 protein family, localized in the cytoplasm and nucleus of a wide range of cells derived from the neural crest [1]. Primarily known for its presence in astrocytes, oligodendrocytes, and Schwann cells, it has been initially recognized as a biomarker for central nervous system (CNS) injuries [2]. Under normal conditions, S100B does not cross the intact blood–brain barrier (BBB). However, when the BBB is compromised due to injury, disease, or inflammation, S100B can leak into the bloodstream. Thus, over the last decade, S100B has emerged as a candidate peripheral biomarker of BBB permeability and dysfunction [3].

Recent research has expanded the understanding of S100B, uncovering its significant role in the pathogenesis of cardiovascular diseases, including chronic heart failure (HF) [4]. Here, S100B has been shown to promote left ventricular remodeling and increase apoptosis in myocardial cells, leading to the deterioration of cardiac function, probably by acting as an intrinsic negative regulator of the myocardial hypertrophy, being involved in the cardiomyocyte response to trophic stimulation [5]. Thus, increased S100B serum levels were associated with the severity of cardiac dysfunction, renal insufficiency, and adverse prognosis in HF [6]. Further, enhanced serum S100B levels were found in patients with either dilated cardiomyopathy or ischemic heart disease [7]. Rat models revealed the ischemic heart as the potential extra cerebral source of S100B [8].

Taken together, recent findings highlight S100B’s dual role as both a cardiac and BBB biomarker. As HF itself can cause BBB breakdown [9], these conditions may be highly interrelated. To better understand the differential role of S100B in the interaction between cardiac disease and CNS, we here provide a comprehensive analysis of S100B in 146 HF patients, who underwent cognitive testing in the context of the Cognition.Matters-HF study. We hypothesized that S100B relates to both cardiac and cognitive dysfunction.

## 2. Results

### 2.1. Patient Characteristics

Sufficient serum for S100B quantification was available from 146 HF patients within the Cognition.Matters-HF study. As shown in Table 1, the median age of the patients was 64 years, with an interquartile range from 56 to 72 years. Females comprised 15% of the sample. The median body mass index (BMI) was 28 kg/m^2^, and systolic and diastolic blood pressures were 135 mmHg and 80 mmHg, respectively. Resting heart rate averaged 63 beats per minute. The median time since heart failure diagnosis was 4 years, with ischemic heart failure present in 66% of the patients. New York Heart Association (NYHA) functional class distribution was 27% in class I, 60% in class II, and 13% in class III. The median 6 min walking distance was 400 m. NT-proBNP levels had a median of 672 pg/mL, and the left ventricular ejection fraction was 44%.

### 2.2. S100B Serum Levels

Median S100B serum levels were 33 (22–47) pg/mL; a histogram is provided in Figure 1A. Patients with S100B levels ≥ 33 pg/mL (median) had a longer duration since heart failure diagnosis, and exhibited higher left atrial volume index compared to those with lower S100B levels (Table 1). Notably, higher S100B levels were associated with a higher prevalence of mild cognitive impairment (MCI; 69% vs. 50%) and lower visual/verbal memory T-scores. No significant differences were observed between the groups regarding other clinical and demographic characteristics, including BMI, blood pressure, heart rate, and the prevalence of comorbid conditions such as diabetes, hypertension, and coronary artery disease.

### 2.3. Association of Clinical Variables with S100B Serum Levels

Table 2 presents the results of univariable and multivariable linear regression models analyzing the association of various clinical and demographic variables with S100B serum levels. In univariable regression analysis, significant associations were observed for NT- proBNP, estimated glomerular filtration rate (eGFR), atrial fibrillation (T-value = 2.04, *p* = 0.043), and angiotensin-converting enzyme inhibitor intake.

When adjusting for multiple variables in the multivariable regression model, NT-proBNP remained significantly associated with S100B serum levels (T-value = 2.63, *p* = 0.010), along with age. The multivariable model explained variance in S100B levels (multiple R^2^ = 0.36). Notably, the associations of sex, BMI, 6 min walking distance, and other clinical parameters with S100B serum levels were not statistically significant in the multivariable analysis, indicating that NT-proBNP is a key independent predictor of S100B levels in this cohort of HF patients (Figure 1B).

### 2.4. Predictors of Mild Cognitive Impairment

More than half of included patients (59%) suffered from MCI, defined as a T-score <40 in at least one of the domains of intensity of attention, visual/verbal memory, and executive function. In the univariable regression analysis (Table 3), significant predictors of MCI included 6 min walking distance (OR (95% CI) 1.00 (0.99–1.00), *p* = 0.014) and smoking history (OR (95% CI) 2.25 (1.11–4.55), *p* = 0.024). In the multivariable regression model, S100B serum levels approached significance (OR (95% CI) 1.03 (1.00–1.06), *p* = 0.083), suggesting a potential association with MCI. Notably, the New York Heart Association (NYHA) functional class (OR (95% CI) 3.05 (1.18–7.88), *p* = 0.021) and left ventricular ejection fraction (OR (95% CI) 1.11 (1.02–1.21), *p* = 0.015) were significant predictors of MCI. Additionally, smoking history remained a significant predictor in the multivariable model (OR (95% CI) 5.59 (1.62–19.22), *p* = 0.006).

Spearman’s correlation revealed significant associations of S100B levels with T-scores of visual/verbal memory (ρ = −0.17; *p* = 0.039; Figure 1C), but not with the intensity of attention (ρ = −0.09; *p* = 0.302) or executive functions (ρ = −0.01; *p* = 0.911). Likewise, NT-proBNP serum levels also correlated with visual–verbal memory (ρ = −0.28; *p* < 0.001), but not with the intensity of attention (ρ = 0.02; *p* = 0.856) or executive functions (ρ = −0.16; *p* = 0.052).

### 2.5. All-Cause Mortality

Lastly, we analyzed whether S100B levels were associated with overall survival. As shown in Figure 2, there was no difference between patients below or above median S100B levels. Cox regression confirmed that S100B had no significant impact both in univariable (HR (95% CI) 1.00 (0.99–1.01); *p* = 0.517) and multivariate (HR (95% CI) 1.01 (1.00–1.03); *p* = 0.142) models, which included the above used clinical confounders.

## 3. Discussion

In this study, we investigated the serum levels of S100B in 146 patients with chronic heart failure (HF) and their association with cardiac and cognitive dysfunction. We found that the median S100B level was 33 pg/mL, with higher levels linked to a longer duration of HF and increased left atrial volume index. Notably, higher S100B levels were associated with a higher prevalence of mild cognitive impairment (MCI) and lower scores in visual/verbal memory tests. Additionally, NT-proBNP levels were significantly correlated with S100B levels, suggesting a potential interaction between heart and brain health in HF patients.

The observed median S100B level of 33 pg/mL in our cohort is somewhat lower than in previous studies involving HF patients. For instance, a small study with 21 HF patients and 21 controls found S100B levels of 51 pg/mL and 17 pg/mL, respectively [7]. Another study with 146 HF patients without renal insufficiency found mean serum levels of 73 pg/mL [6]. Interestingly, these patients had high serum NT-proBNP levels of 2881 ± 3454.2 pg/mL, which might explain higher S100B levels compared to our cohort with a relatively mild HF phenotype. In a third study, patients with ischemic HF (*n* = 124) and dilated HF (*n* = 91) had S100B plasma levels of ~90 pg/mL, compared to ~70 pg/mL in controls (*n* = 215) [10]. It must be noted that variations in assay sensitivity, specificity, and calibration standards can significantly impact the measured levels.

Our findings demonstrated a significant correlation between S100B and NT-proBNP levels (T-value = 2.63, *p* = 0.010), even in multivariable models. On the one hand, elevated S100B levels can result from myocardial stress or damage, particularly under ischemic conditions, where cardiomyocytes express and release S100B [8]. In that study, both S100B and troponin T levels increased significantly after 20 min of ischemia, suggesting that the intracellular protein S100B is released from cardiomyocytes into serum during cardiac injury. While we did not explicitly quantify troponin levels, the strong correlation of S100B with NT-proBNP, a marker for myocardial stress, suggests S100B release also upon chronic myocardial dysfunction. On the other hand, HF itself can lead to cerebral hypoperfusion and systemic inflammation [11], compromising BBB integrity and allowing S100B from astrocytes to enter the bloodstream. This mechanism is supported by the association of elevated S100B levels with cognitive impairment in our cohort. Taken together, elevated S100B levels in HF patients likely reflect both cardiac stress and BBB disruption, highlighting the interconnection between cardiac and neurological health in HF. This dual role of S100B enhances its value as a biomarker for comprehensive disease assessment.

Higher S100B levels were associated with lower visual/verbal memory scores and a higher prevalence of MCI. This finding aligns with the hypothesis that HF-related neuroinflammation and BBB disruption may contribute to cognitive decline [12]. In the field of neurology, S100B serum levels are related to cognitive dysfunction in patients with, e.g., cerebral infarction [13], sepsis-associated encephalopathy [14], or cardiac arrest [15]. We here expand these interrelations to an HF cohort. Interestingly, transiently elevated levels of S100B are suggestive of minor acute cerebral damage in the first days following MI and are associated with depressive symptoms in the year following MI [16] and in patients with end-stage renal disease [17].

Despite the significant correlation between S100B and NT-proBNP levels, our study found that S100B had no impact on mortality. This is a little in contrast with a former HF study, claiming that increased S100B levels were associated with major cardiac events, which was defined as all-cause death, requirement for heart transplantation, and refractory HF requiring multiple hospitalizations [6]. Several factors might explain this finding: First, mortality in HF is influenced by a multitude of factors, including the severity of cardiac dysfunction, comorbid conditions, and treatment adherence. While S100B reflects myocardial stress and potential BBB disruption, it may not capture the full spectrum of prognostic factors that contribute to mortality risk in HF. Second, S100B levels may be more indicative of acute injury or stress rather than long-term outcomes. Elevated S100B can signal acute myocardial damage or neuroinflammation, but chronic mortality risk in HF patients might be more closely associated with other markers like NT-proBNP [18], which reflects sustained ventricular stress and volume overload. Third, the relatively mild HF phenotype in our cohort, with moderate NT-proBNP levels, may limit the power to detect an association between S100B and mortality. In more severe HF cases, S100B might show a stronger relationship with adverse outcomes.

Our findings also highlight the need for careful consideration of other potential sources of elevated S100B levels. In this regard, analysis of healthy human organ samples by Western blot revealed high S100B levels of about 100 ng/mg brain tissue, while peripheral tissues (including adipose tissue) had levels below 10 ng/mL, suggesting that extracranial sources of S100B do not majorly affect serum levels [19]. However, myocardial tissue was not analyzed in this study. 

Serum levels of S100B are standard in monitoring advanced malignant melanoma patients. Here, it is well known that obesity, liver cirrhosis, migraine, chronic kidney disease, and previous stroke associated with false-positive S100B levels [20]. Regarding obesity, previous reports have shown conflicting results regarding the correlation of BMI with S100B [19,21,22,23,24], which could not be confirmed in our cohort. Regarding renal dysfunction, our findings confirm a univariate association between S100B and eGFR, which was also described by others [6,21,25]. In critically ill patients, serum levels of S100B were found to be elevated, even in the absence of apparent brain damage, and correlated positively with lactate levels and negatively with mean arterial pressure, pH, and hemoglobin, suggesting that serum S100B protein concentration may be related to peripheral tissue hypoperfusion [26]. The associations with blood pressure and hemoglobin could not be confirmed in our study. 

This study is pioneering in linking serum S100B levels with MCI in HF patients. The long follow-up period of 10 years and comprehensive cognitive testing are significant strengths. Additionally, the comprehensive clinical data collected enabled the use of robust multivariate models to adjust for potential confounders, enhancing the validity of the findings. However, the study has several limitations. The cross-sectional nature of the data limits the ability to infer causality between S100B levels and MCI. Furthermore, the BBB function was not explicitly investigated in our study. Future studies using e.g., contrast-enhanced MRI are needed to consolidate our findings. The relatively small sample size and a lack of a control cohort may reduce the power to detect stronger associations and limit the generalizability of the findings. Furthermore, some associations observed were relatively weak, which may impact the clinical applicability of S100B as a biomarker in this context. Also, the fact that patients were included between 2011 and 2014 explains why none of the patients received sodium glucose-linked transporter 2 or sacubitril/valsartan, as these substances were not approved for the German market at that time.

In conclusion, our study demonstrates that elevated serum S100B levels in chronic heart failure (HF) patients are associated with both cardiac and cognitive dysfunction. Notably, higher S100B levels correlated with a higher prevalence of MCI and lower visual/verbal memory scores. The significant correlation between S100B and NT-proBNP levels suggests an interaction between cardiac stress and brain health in HF. However, S100B did not impact mortality, likely due to its reflection of acute injury rather than long-term outcomes and the mild HF phenotype of our cohort. These findings highlight the dual role of S100B as a marker of both myocardial stress and BBB disruption, necessitating a comprehensive assessment of elevated S100B levels in clinical practice. Future studies should aim to include a broader panel of biomarkers to validate and expand upon our findings to offer a more robust predictive model for patient outcomes, especially concerning long-term survival and disease progression.

## 4. Materials and Methods

### 4.1. Study Design

The Cognition.Matters-HF study is a prospective, investigator-initiated, monocentric follow-up study. The study protocol received approval from the local ethics committee and adheres to the principles outlined in the Declaration of Helsinki. All participants provided written informed consent. Eligibility criteria included adult patients with clinically confirmed HF, as defined by the guidelines of the European Society of Cardiology at the time of study entry [27,28]. Eligible patients were consecutively recruited in the outpatient clinic at the Comprehensive Heart Failure Center Würzburg and included between the years 2011 and 2014. Echocardiography was used as the initial screening procedure. Patients with new-onset or acutely decompensated HF were excluded. Additional exclusion criteria included a history of clinical stroke, evident psychiatric disorders (including depression or dementia), carotid artery stenosis greater than 50%, or any implants or devices that would interfere with brain magnetic resonance imaging. Echocardiography was utilized as the initial screening method. Baseline investigations, along with follow-up examinations at 12, 36, and 60 months, included thorough evaluations by a cardiologist, neuropsychologist, neurologist, and neuroradiologist. Efforts were made to ensure all diagnostic procedures during each visit were completed within two days. Clinical examination, electrocardiography, echocardiography, 24 h Holter electrocardiography, 24 h blood pressure measurement, and the 6 min walk test (6-MWT) were conducted following standard operating procedures at the Comprehensive Heart Failure Center Würzburg. 

### 4.2. S100B Measurements

We collected non-fasting venous blood samples from all patients for routine clinical chemistry investigations at the certified facility of the University Hospital Würzburg. Participants were seated for at least 5 min before the blood draw. Serum samples were processed immediately. They were kept at room temperature for 30 min and then centrifuged for 10 min at 2000× *g*. The serum was aliquoted into dedicated fluid tissue tubes (Micronic, Lelystad, The Netherlands) and stored in a standardized interdisciplinary biomaterial bank at −80 °C until analysis [29]. Serum levels of S100B were quantified using the Human S100B ELISA kit (EZHS100B-33K) from Millipore (Billerica, MA, USA). This sandwich enzyme immunoassay involved adding serum samples, standards, and controls to microplate wells pre-coated with S100B-specific antibodies. Following incubation with a biotinylated detection antibody and an avidin-horseradish peroxidase (HRP) conjugate, a substrate solution was added, resulting in a color change proportional to the S100B concentration. The color intensity was measured spectrophotometrically. 

The reproducibility of S100B measurements was ensured through rigorous adherence to standardized protocols and quality control procedures. The Human S100B ELISA kit (EZHS100B-33K, Millipore, Billerica, MA, USA) employed for quantification is a well-established, validated method with high sensitivity and specificity for detecting S100B levels. Intra-assay and inter-assay variability were minimized by performing all assays in duplicate and including controls with known S100B concentrations in each run. The intra-assay coefficient of variation (CV) was consistently below 10%, and the inter-assay CV was maintained below 15%, ensuring reliable and reproducible results across different samples and assay runs. Additionally, all measurements were performed by trained personnel following the manufacturer’s instructions and laboratory best practices.

### 4.3. Neuropsychological Testing

Extensive neuropsychological testing included the test battery of attentional performance for the domain intensity and selectivity of attention. The visual and verbal memory test was used to test performance in the domain of visual/verbal memory, while working memory was assessed using the digit span forward and block tapping span forward tests. The Regensburger word fluency test and HAMASCH 5-point-test tested visual/verbal fluency. These tests have been validated in healthy volunteers to account for the modifying effect of age, sex, and educational level. These factors were considered in the T-standardized output values with a mean of 50 and a standard deviation of 10 as the mean reference comparator. Executive function T-values are calculated by the mean of selectivity of attention, working memory, and visual/verbal fluency.

### 4.4. Statistical Considerations

Data are presented as median (interquartile range) or count (percentage), as appropriate. Statistical analyses included *t*-tests for comparing means of normally distributed variables and Mann–Whitney tests for non-normally distributed variables. Chi-squared and Fisher’s exact tests were used for analyzing categorical data. Univariable and multivariable regression analyses were performed to identify predictors of outcomes, adjusting for potential confounders. Spearman’s rho correlation was used to assess the strength and direction of associations between non-parametric variables. Additionally, Cox regression analyses were conducted to evaluate the impact of various predictors on survival outcomes, providing hazard ratios and corresponding *p*-values. Kaplan–Meier plots were generated to visualize survival distributions, with log-rank test *p*-values reported to compare differences between groups.

## Figures and Tables

**Figure 1 ijms-25-09094-f001:**
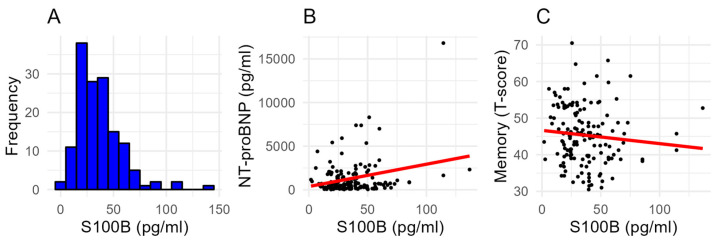
S100B serum levels. (**A**) Histogram of serum S100B levels. (**B**) Correlation plot of serum S100B serum levels with serum NT-proBNP levels. (**C**) Correlation plot of serum S100B serum levels with performance in visual/verbal memory testing (T-scores).

**Figure 2 ijms-25-09094-f002:**
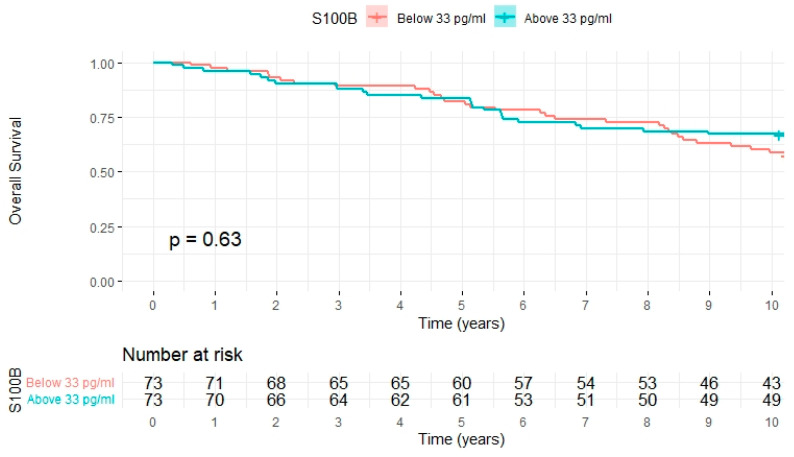
Kaplan–Meier plot for overall survival according to S100B level.

**Table 1 ijms-25-09094-t001:** Patient characteristics according to median S100B serum levels.

	Full Sample (*n* = 146)	S100B < 33 pg/mL (*n* = 73)	S100B ≥ 33 pg/mL (*n* = 73)	*p* Value (Two-Sided)
Age (years)	64 (56–72)	63 (54–71)	65 (58–74)	0.097
Female sex	22 (15)	12 (16)	10 (14)	0.644
Body mass index (kg/m^2^)	28 (26–32)	28 (26–32)	29 (25–32)	0.813
Systolic blood pressure (mmHg)	135 (124–151)	137 (124–150)	135 (123–153)	0.423
Diastolic blood pressure (mmHg)	80 (75–89)	81 (75–90)	80 (75–88)	0.455
Resting heart rate (1/min)	63 (58–70)	64 (58–71)	62 (57–70)	0.151
Time since diagnosis of heart failure (years)	4 (1–10)	3 (1–7)	5 (2–12)	0.014
Ischemic heart failure	96 (66)	48 (66)	48 (66)	>0.999
New York Heart Association functional class I	39 (27)	19 (26)	20 (27)	0.852
New York Heart Association functional class II	88 (60)	47 (64)	41 (56)	0.310
New York Heart Association functional class III	19 (13)	7 (10)	12 (16)	0.219
6 min walking distance (m)	400 (340–460)	380 (340–460)	420 (350–460)	0.224
NT-proBNP (pg/mL)	672 (237–1736)	632 (219–1475)	703 (265–1926)	0.063
Left ventricular ejection fraction (%)	44 (38–48)	43 (38–48)	44 (38–49)	0.241
Heart failure with preserved ejection fraction	24 (16)	9 (12)	15 (21)	0.180
Heart failure with midrange ejection fraction	72 (49)	41 (56)	31 (43)	0.098
Heart failure with reduced ejection fraction	50 (34)	23 (32)	27 (37)	0.485
Left atrial volume index (mL/m^2^)	38 (29–52)	38 (29–48)	39 (31–55)	0.041
Left ventricular end-diastolic volume (mL)	167 (110–162)	127 (112–166)	124 (110–153)	0.244
E/e’ ratio	10 (8–13)	10 (8–12)	10 (7–13)	0.872
Deceleration time (ms)	188 (146–231)	195 (159–241)	172 (144–221)	0.069
Arterial hypertension	116 (80)	54 (74)	62 (85)	0.101
Diabetes mellitus	42 (29)	24 (33)	18 (25)	0.273
Hyperlipidemia	105 (72)	53 (73)	52 (71)	0.854
Smoking (ever)	88 (60)	44 (60)	44 (60)	>0.999
Coronary artery disease	100 (69)	51 (70)	49 (67)	0.722
History of myocardial infarction	80 (55)	42 (58)	38 (55)	0.506
Atrial fibrillation	33 (23)	14 (19)	19 (26)	0.322
Angiotensin-converting enzyme inhibitor	86 (59)	48 (66)	38 (52)	0.093
Angiotensin-1 receptor antagonist	48 (33)	19 (26)	29 (40)	0.078
Sacubitril/valsartan	0 (0)	0 (0)	0 (0)	>0.999
Beta-blocker	131 (90)	65 (89)	66 (90)	0.785
Mineralocorticoid receptor antagonist	54 (37)	26 (36)	28 (38)	0.732
Sodium glucose linked transporter 2 inhibitor	0 (0)	0 (0)	0 (0)	>0.999
Statins	101 (69)	53 (73)	48 (66)	0.370
Ezetimibe	9 (6)	4 (6)	5 (7)	>0.999
Loop diuretic	80 (55)	36 (49)	44 (60)	0.183
Thiazide	23 (16)	9 (12)	14 (19)	0.256
Estimated glomerular filtration rate (mL/min/1.73 m^2^)	68 (55–80)	74 (59–84)	61 (48–76)	0.009
Alanine transferase (U/L)	24 (19–34)	23 (19–34)	25 (18–34)	0.749
Aspartate transferase (U/L)	26 (21–31)	26 (21–31)	25 (22–31)	0.702
Albumin (g/dL)	4.6 (4.4–4.7)	4.6 (4.4–4.8)	4.6 (4.4–4.7)	0.466
Creatine kinase (U/L)	102 (75–153)	95 (70–139)	113 (77–168)	0.163
C-reactive protein (mg/dL)	0.2 (0.1–0.5)	0.2 (0.1–0.6)	0.2 (0.1–0.4)	0.139
Hemoglobin (g/dL)	14.5 (13.6–15.3)	14.3 (13.6–15.4)	14.5 (13.7–15.2)	0.804
Chronic kidney disease G1	15 (10)	7 (10)	8 (11)	0.785
Chronic kidney disease G2	79 (54)	47 (64)	32 (44)	0.013
Chronic kidney disease G3	47 (32)	19 (26)	28 (38)	0.111
Chronic kidney disease G4	5 (3)	0 (0)	5 (7)	0.058
Chronic kidney disease G5	0 (0)	0 (0)	0 (0)	>0.999
Mild cognitive impairment	86 (59)	36 (50)	50 (69)	0.023
Intensity of attention (T-score)	42 (37–47)	42 (38–46)	40 (35–47)	0.152
Visual/verbal memory (T-score)	45 (39–51)	46 (41–53)	43 (39–48)	0.006
Executive functions (T-score)	45 (42–50)	45 (41–50)	45 (42–50)	0.469

**Table 2 ijms-25-09094-t002:** Uni- and multivariable linear regression models of S100B serum levels.

S100B Serum Level (pg/mL)	Univariable Regression			Multivariable Regression	
	R^2^	T-Value	*p* Value	T-Value	*p* Value
Age (years)	0.01	1.43	0.155	2.02	0.046
Sex (female)	0.00	-0.46	0.646	1.18	0.243
Body mass index (kg/m^2^)	0.01	1.33	0.186	1.17	0.244
6 min walking distance (m)	0.01	1.03	0.303	1.40	0.164
New York Heart Association functional class	0.00	−0.01	0.988	−0.73	0.466
NT-proBNP (pg/mL)	0.07	3.34	0.001	2.27	0.026
Left ventricular ejection fraction (%)	0.01	0.96	0.339	0.70	0.486
Left atrial volume index (mL/m^2^)	0.01	0.96	0.340	−1.04	0.299
Left ventricular end-diastolic volume (mL)	0.01	−1.04	0.299	−0.83	0.410
E/e’ ratio	0.00	0.07	0.947	−0.35	0.730
Deceleration time (ms)	0.01	−1.01	0.312	−1.08	0.281
Estimated glomerular filtration rate (mL/min/1.73 m^2^)	0.07	−3.29	0.001	−1.90	0.061
Alanine transferase (U/L)	0.00	0.20	0.842	1.81	0.074
Aspartate transferase (U/L)	0.00	0.53	0.596	−1.30	0.198
Albumin (g/dL)	0.01	−1.06	0.291	−1.09	0.280
Creatine kinase (U/L)	0.02	1.69	0.094	0.94	0.350
C-reactive protein (mg/dL)	0.01	−0.86	0.389	−0.76	0.450
Hemoglobin (g/dL)	0.00	−0.16	0.870	0.97	0.332
Arterial hypertension	0.02	−1.62	0.108	−1.25	0.214
Diabetes mellitus	0.01	1.07	0.288	0.64	0.524
Hyperlipidemia	0.00	0.06	0.954	−0.38	0.705
Smoking (ever)	0.00	−0.23	0.817	−1.89	0.062
Coronary artery disease	0.01	1.08	0.283	0.13	0.900
History of myocardial infarction	0.01	1.05	0.294	−0.51	0.612
Atrial fibrillation	0.03	2.04	0.043	0.84	0.400
Angiotensin-converting enzyme inhibitor	0.03	2.23	0.027	0.60	0.552
Angiotensin-1 receptor antagonist	0.02	−1.50	0.135	−0.53	0.597
Beta-blocker	0.00	−0.85	0.397	−1.31	0.192
Mineralocorticoid receptor antagonist	0.00	−0.57	0.567	−0.78	0.440
Statins	0.01	1.35	0.178	1.89	0.062
Ezetimibe	0.00	0.56	0.577	−0.23	0.819
Loop diuretic	0.02	−1.89	0.061	−0.47	0.640
Thiazide	0.01	−1.09	0.278	−0.95	0.346

**Table 3 ijms-25-09094-t003:** Uni- and multivariable logistic regression models of mild cognitive impairment.

Mild Cognitive Impairment	Univariable Regression		Multivariable Regression	
	OR (95% CI)	*p* Value	OR (95% CI)	*p* Value
S100B (pg/mL)	1.01 (0.99–1.03)	0.213	1.03 (1.00–1.06)	0.083
Age (years)	1.03 (1.00–1.06)	0.073	0.98 (0.92–1.04)	0.474
Sex (female)	2.02 (0.74–5.51)	0.170	0.38 (0.06–2.59)	0.326
Body mass index (kg/m^2^)	0.97 (0.91–1.04)	0.414	0.94 (0.83–1.06)	0.280
6 min walking distance (m)	1.00 (0.99–1.00)	0.014	1.00 (0.99–1.00)	0.389
New York Heart Association functional class	1.56 (0.90–2.71)	0.111	3.05 (1.18–7.88)	0.021
NT-proBNP (pg/mL)	1.00 (1.00–1.00)	0.486	1.00 (1.00–1.00)	0.463
Left ventricular ejection fraction (%)	1.02 (0.98–1.06)	0.333	1.11 (1.02–1.21)	0.015
Left atrial volume index (mL/m^2^)	1.01 (0.99–1.03)	0.431	1.00 (0.96–1.04)	0.873
Left ventricular end-diastolic volume (mL)	1.00 (0.99–1.01)	0.531	1.01 (0.99–1.02)	0.392
E/e’ ratio	0.97 (0.89–1.04)	0.386	0.94 (0.84–1.06)	0.343
Deceleration time (ms)	1.00 (0.99–1.00)	0.185	0.99 (0.98–1.00)	0.229
Estimated glomerular filtration rate (mL/min/1.73 m^2^)	0.99 (0.97–1.00)	0.122	1.00 (0.97–1.03)	0.991
Alanine transferase (U/L)	0.99 (0.96–1.01)	0.272	0.98 (0.93–1.03)	0.434
Aspartate transferase (U/L)	1.01 (0.97–1.04)	0.769	1.05 (0.98–1.13)	0.161
Albumin (g/dL)	0.48 (0.13–1.76)	0.269	2.26 (0.22–23.30)	0.493
Creatine kinase (U/L)	1.00 (0.99–1.00)	0.637	1.00 (0.99–1.00)	0.299
C-reactive protein (mg/dL)	1.09 (0.72–1.67)	0.674	1.35 (0.70–2.61)	0.373
Hemoglobin (g/dL)	0.90 (0.71–1.14)	0.381	0.73 (0.45–1.20)	0.212
Arterial hypertension	0.68 (0.30–1.54)	0.354	1.42 (0.37–5.47)	0.609
Diabetes mellitus	0.86 (0.41–1.79)	0.685	0.51 (0.14–1.77)	0.286
Hyperlipidemia	1.62 (0.75–3.48)	0.217	1.40 (0.32–6.15)	0.657
Smoking (ever)	2.25 (1.11–4.55)	0.024	5.59 (1.62–19.22)	0.006
Coronary artery disease	0.85 (0.42–1.72)	0.641	0.16 (0.02–1.30)	0.086
History of myocardial infarction	1.10 (0.57–2.15)	0.772	3.07 (0.57–16.49)	0.191
Atrial fibrillation	1.07 (0.49–2.37)	0.863	1.19 (0.30–4.70)	0.799
Angiotensin-converting enzyme inhibitor	1.18 (0.60–2.32)	0.628	5.13 (0.76–34.54)	0.093
Angiotensin-1 receptor antagonist	1.21 (0.60–2.43)	0.598	6.97 (0.85–57.26)	0.071
Beta-blocker	0.76 (0.26–2.23)	0.620	0.41 (0.09–1.82)	0.242
Mineralocorticoid receptor antagonist	0.89 (0.45–1.77)	0.734	0.90 (0.31–2.61)	0.847
Statins	1.30 (0.63–2.69)	0.484	1.08 (0.15–7.93)	0.938
Ezetimibe	2.56 (0.59–11.16)	0.210	6.28 (0.74–53.54)	0.093
Loop diuretic	0.66 (0.34–1.29)	0.228	0.77 (0.24–2.45)	0.655
Thiazide	0.93 (0.37–2.30)	0.868	0.69 (0.16–2.90)	0.614

## Data Availability

The data presented in this study are available on request from the corresponding author.

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
