# Peer review of "S100B Serum Levels in Chronic Heart Failure Patients: A Multifaceted Biomarker Linking Cardiac and Cognitive Dysfunction"

_ijms, 2024, doi:10.3390/ijms25169094_

Round 1

Reviewer 1 Report

Comments and Suggestions for Authors

In this manuscript (ID# ijms-3155392) entitled “S100B Serum Levels in Chronic Heart Failure Patients: A Multifaceted Biomarker Linking Cardiac and Cognitive Dysfunction”, the authors, Traub et al, studied the S100B serum levels in chronic heart failure (HF) patients. Their results have demonstrated that Serum S100B were passively co-related with HF duration, left atrial volume index, mild cognitive impairment, NT-proBNP levels, and memory scores. They conclude that elevated serum S100B in CHF patients are associated with both cardiac and cognitive dysfunction. However, there are several major concerns, which are listed in the following paragraphs:

1. Similar previous studies have been done. What is the aim of this study to make this manuscript distinctive?

2.  BBB disruption is mentioned in the manuscript several times; however, the BBB function was not examined in the current study.

3. There is no healthy control to be compared with chronic heart failure patients. This impair the evidence supporting for their conclusion.

4. Recently studies have demonstrated that S100B protein is expressed not only in the central nervous system (CNS), but also in peripheral tissues, such as heart, adipose. Is it possible that the intracellular protein S100B is released from cardiomyocytes into serum during cardiac injury?

5. Is cardiac dysfunction also corelated with memory score?

Comments on the Quality of English Language

English language is fine. Only minor editing is required.

Author Response

Reviewer #1:

In this manuscript (ID# ijms-3155392) entitled “S100B Serum Levels in Chronic Heart Failure Patients: A Multifaceted Biomarker Linking Cardiac and Cognitive Dysfunction”, the authors, Traub et al, studied the S100B serum levels in chronic heart failure (HF) patients. Their results have demonstrated that Serum S100B were passively co-related with HF duration, left atrial volume index, mild cognitive impairment, NT-proBNP levels, and memory scores. They conclude that elevated serum S100B in CHF patients are associated with both cardiac and cognitive dysfunction.

             The authors want to thank the reviewer for this precise summary of the study.

However, there are several major concerns, which are listed in the following paragraphs:

  1. Similar previous studies have been done. What is the aim of this study to make this manuscript distinctive?

Thanks for this important remark. Indeed, as discussed in the manuscript, previous studies with comparable sample sizes have quantified S100B serum levels in chronic heart failure (HF) patients (PMIDs 18068619, 21640093, 32590820). However, the following points make the current study distinctive:

  1. Other HF studies did not use multivariate models to reveal independent relations of S100B levels with NT-proBNP levels, which is important as multiple factors (e.g. renal function) can affect both.
  2. The presence of cognitive testing in our study is unique to other HF cohorts. This allowed for testing whether S100B also related to cognitive dysfunction in a cohort of HF patients.
  3. The long follow-up of 10 years allowed for assessing the relevance of altered S100B serum levels regarding all-cause mortality.

Accordingly, we have now updated the discussion of the revised manuscript:

“This study is pioneering in linking serum S100B levels with MCI in HF patients. Long follow-up period of 10 years and comprehensive cognitive testing are significant strengths. Additionally, the comprehensive clinical data collected enabled the use of robust multi-variate models to adjust for potential confounders, enhancing the validity of the findings.

  1. BBB disruption is mentioned in the manuscript several times; however, the BBB function was not examined in the current study.

This is a valid remark. The authors are very aware that elevated S100B may only be a hint for BBB disruption, as discussed. We have added these thoughts to the discussion of the revised manuscript:

“Furthermore, BBB function was not explicitly investigated in our study. Future studies using e.g. contrast-enhanced MRI are needed to consolidate our findings.”

  1. There is no healthy control to be compared with chronic heart failure patients. This impair the evidence supporting for their conclusion.

The authors are aware of this limitation. Other studies have already demonstrated that S100B levels are elevated in HF compared to healthy controls (PMIDs 18068619, 21640093, 32590820). As the levels in our HF cohort were comparable with these studies, we focused on determinants of S100B levels and its impact on cognition and mortality within the HF population. We have now added this limitation to the discussion of the revised manuscript:

“The relatively small sample size and a lack of a control cohort may reduce the power to detect stronger associations and limit the generalizability of the findings.”

  1. Recently studies have demonstrated that S100B protein is expressed not only in the central nervous system (CNS), but also in peripheral tissues, such as heart, adipose. Is it possible that the intracellular protein S100B is released from cardiomyocytes into serum during cardiac injury?

The reviewer discusses the probably most important point of this paper. There is indeed a lot of evidence for extra-cerebral sources of S100B. This study focused on HF and cardiomyocytes as possible source of S100B, as shown e.g. in isolated ischemic rat hearts (PMID 15921704). In this study, both S100B and troponin T levels increased significantly after 20 minutes of ischemia, suggesting that the intracellular protein S100B is released from cardiomyocytes into serum during cardiac injury. We did not quantify troponin in our cohort, but it is feasible that cardiac injury (which also occurs in chronic conditions like HF) may be responsible for S100B release upon higher NT-proBNP levels. We included these thought into the discussion:

In that study, both S100B and troponin T levels increased significantly after 20 minutes of ischemia, suggesting that the intracellular protein S100B is released from cardiomyocytes into serum during cardiac injury. While we did not explicitly quantify troponin levels, the strong correlation of S100B with NT-proBNP, a marker for myocardial stress, suggests S100B release also upon chronic myocardial dysfunction.”

  1. Is cardiac dysfunction also correlated with memory score?

Thanks for raising this important point. Paralleling the correlation analysis of S100B levels with distinct cognitive domains (including memory), we have now added a few lines on associations of NT-proBNP with these domains:

“Likewise, NT-proBNP serum levels also correlated with visual verbal memory (ρ = -0.28; p < 0.001), but not with intensity of attention (ρ = 0.02; p = 0.856) or executive functions (ρ = -0.16; p = 0.052).”

Reviewer 2 Report

Comments and Suggestions for Authors

To:

Editorial Board

Title: “S100B Serum Levels in Chronic Heart Failure Patients: A Multifaceted Biomarker Linking Cardiac and Cognitive Dysfunction”

Dear Editor,

I read this interesting paper and I think that some points should be addressed:

-       Authors should include a flow chart of the study. Were patients consecutively included?

-       Reproducibility of the SB100 measurements should be declared. Please provide.

-       There is no mention of antilipemic drugs. The impact of statins and other lipid-lowering drugs should be discussed and table 1 should be updated with data.

-       Sacubitril/valsartan and SGLT2i were not administered to these patients. Why? How did authors update therapy for HF? These questions are fundamental as lack of optimized therapy for HF might impact on final outcomes and results.

-       Authors should divide patients in relation to the type of HF: HFrEF, HFmrEF, and HFpEF. These categories are different each other and had different impact in outcomes. Please revise the paper.

-       Half of patients were on loop diuretics, while about one third on MRA. Can the authors explain these data? This seems a lack of therapy optimization. Please discuss such a point.

-       Please include laboratory data in table 1. How many patients had kidney failure? What degree? Please discuss and update table.

-       The multivariate regression analysis should include ALL of these data in order to evaluate the impact of confounding factors on final results.

Author Response

Reviewer #2:

I read this interesting paper and I think that some points should be addressed:

             Thanks for the positive feedback on this paper.

Authors should include a flow chart of the study. Were patients consecutively included?

As published earlier (PMID 29885954), patients were indeed consecutively recruited in the outpatient clinic at the Comprehensive Heart Failure Center Würzburg and included between the years 2011 and 2014. Echocardiography was used as the initial screening procedure. Care was taken to complete all diagnostic procedures per visit within 2 days. We added this information to the methods section and think that no flow-chart is needed here, as follow-up data was not used in this investigation:

“Eligible patients were consecutively recruited in the outpatient clinic at the Comprehensive Heart Failure Center Würzburg and included between the years 2011 and 2014. Echocardiography was used as the initial screening procedure.”

Reproducibility of the S100B measurements should be declared. Please provide.

We agree that more information on the reproducibility of the S100B measurements must be provided for the interested reader. Thus, we added the following lines to the methods section:

“The reproducibility of S100B measurements was ensured through rigorous adherence to standardized protocols and quality control procedures. The Human S100B ELISA kit (EZHS100B-33K, Millipore, Billerica, MA, USA) employed for quantification is a well-established, validated method with high sensitivity and specificity for detecting S100B levels. Intra-assay and inter-assay variability were minimized by performing all assays in duplicate and including controls with known S100B concentrations in each run. The intra-assay coefficient of variation (CV) was consistently below 10%, and the inter-assay CV was maintained below 15%, ensuring reliable and reproducible results across different samples and assay runs. Additionally, all measurements were performed by trained personnel following the manufacturer's instructions and laboratory best practices.”

There is no mention of antilipemic drugs. The impact of statins and other lipid-lowering drugs should be discussed and table 1 should be updated with data.

The authors thank the reviewer for this important aspect. There was no difference of patient frequencies taking statins or ezetimibe between S100B groups. We have now added these data to Table 1:

full sample

(n=146)

S100B < 33pg/ml

(n=73)

S100B ≥ 33pg/ml

(n=73)

p value

(two-sided)

Statins

101 (69)

53 (73)

48 (66)

0.370

Ezetimibe

9 (6)

4 (6)

5 (7)

>0.999

Sacubitril/valsartan and SGLT2i were not administered to these patients. Why? How did authors update therapy for HF? These questions are fundamental as lack of optimized therapy for HF might impact on final outcomes and results.

Thanks for this mindful remark. As discussed above, the recruitment of the Cognition.Matters-HF study (2011-2014) took place before sodium glucose linked transporter 2 (SGLT2) and sacubitril/valsartan were approved for the German market. This explains why almost no patient received these drugs. During the conduction of this study we aimed to treat patients according to the then-current guidelines of the European Society of Cardiology (ESC Guidelines 2008/2012; PMIDs 18826876, 22611136). We have now clarified this in the methods and discussion section accordingly:

“Eligibility criteria included adult patients with clinically confirmed HF, as defined by the guidelines of the European Society of Cardiology at the time of study entry [27,28]”

“Also, the fact that patients were include between 2011 and 2014 explains why none of the patients received sodium glucose linked transporter 2 or sacubitril/valsartan, as these substances were not approved for the German market at that time.”

Authors should divide patients in relation to the type of HF: HFrEF, HFmrEF, and HFpEF. These categories are different each other and had different impact in outcomes. Please revise the paper.

This is an interesting thought. Indeed, these categories (which base on left ventricular ejection fraction) might influence our findings. However, this aspect is (at least party) already covered by the continuous variable left ventricular ejection fraction itself, which is already included in the multivariate regression models.  Nevertheless, we have now added information on the frequencies of these categories to Table 1 – with no difference in their frequencies:

full sample

(n=146)

S100B < 33pg/ml

(n=73)

S100B ≥ 33pg/ml

(n=73)

p value

(two-sided)

Heart failure with preserved ejection fraction

24 (16)

9 (12)

15 (21)

0.180

Heart failure with midrange ejection fraction

72 (49)

41 (56)

31 (43)

0.098

Heart failure with reduced ejection fraction

50 (34)

23 (32)

27 (37)

0.485

Half of patients were on loop diuretics, while about one third on MRA. Can the authors explain these data? This seems a lack of therapy optimization. Please discuss such a point.

As stated above, patients were recruited between 2011 and 2014. In most cases, at baseline visit, it was the first time for the patients to visit our outpatient clinic at the German Comprehensive Heart Failure Center. This might explain, why only one third had received mineralocorticoid receptor antagonist, suggesting a previous lack of therapy optimization indeed. During the conduction of this study (after baseline investigation), we aimed to treat patients according to the then-current guidelines of the European Society of Cardiology (ESC Guidelines 2008/2012; PMIDs 18826876, 22611136).

Please include laboratory data in table 1. How many patients had kidney failure? What degree? Please discuss and update table.

Following this constructive feedback of this reviewer, we have now included laboratory parameters including data on chronic kidney failure into Table 1. Of note, data on albuminuria was not available:

full sample

(n=146)

S100B < 33pg/ml

(n=73)

S100B ≥ 33pg/ml

(n=73)

p value

(two-sided)

Estimated glomerular filtration rate (ml/min/1.73m²)

68 (55-80)

74 (59-84)

61 (48-76)

0.009

Alanine transferase (U/l)

24 (19-34)

23 (19-34)

25 (18-34)

0.749

Aspartate transferase (U/l)

26 (21-31)

26 (21-31)

25 (22-31)

0.702

Albumin (g/dl)

4.6 (4.4-4.7)

4.6 (4.4-4.8)

4.6 (4.4-4.7)

0.466

Creatine kinase (U/l)

102 (75-153)

95 (70-139)

113 (77-168)

0.163

C-reactive protein (mg/dl)

0.2 (0.1-0.5)

0.2 (0.1-0.6)

0.2 (0.1-0.4)

0.139

Hemoglobin (g/dl)

14.5 (13.6-15.3)

14.3 (13.6-15.4)

14.5 (13.7-15.2)

0.804

Chronic kidney disease G1

15 (10)

7 (10)

8 (11)

0.785

Chronic kidney disease G2

79 (54)

47 (64)

32 (44)

0.013

Chronic kidney disease G3

47 (32)

19 (26)

28 (38)

0.111

Chronic kidney disease G4

5 (3)

0 (0)

5 (7)

0.058

Chronic kidney disease G5

0 (0)

0 (0)

0 (0)

>0.999

The multivariate regression analysis should include ALL of these data in order to evaluate the impact of confounding factors on final results.

Thanks for this suggestion. As laboratory data was already included, we have now added data on medication to the multivariate models of the revised manuscript (Tables 2+3). Here, intake of Angiotensin-converting enzyme inhibitor associated with S100B levels in the univariable models, but not in multivariable models. No other significant associations were found and it not impact the results in general. Numbers in the text were updated accordingly.

S100B serum level (pg/ml)

Univariable regression

Multivariable regression*

T-value

p value

T-value

p value

Angiotensin-converting enzyme inhibitor

0.03

2.23

0.027

0.60

0.552

Angiotensin-1 receptor antagonist

0.02

-1.50

0.135

-0.53

0.597

Beta-blocker

0.00

-0.85

0.397

-1.31

0.192

Mineralocorticoid receptor antagonist

0.00

-0.57

0.567

-0.78

0.440

Statins

0.01

1.35

0.178

1.89

0.062

Ezetimibe

0.00

0.56

0.577

-0.23

0.819

Loop diuretic

0.02

-1.89

0.061

-0.47

0.640

Thiazide

0.01

-1.09

0.278

-0.95

0.346

Mild cognitive impairment

Univariable regression

Multivariable regression

OR (95% CI)

p value

OR (95% CI)

p value

S100B (pg/ml)

1.01 (0.99-1.03)

0.213

1.03 (1.00-1.06)

0.083

Age (years)

1.03 (1.00-1.06)

0.073

0.98 (0.92-1.04)

0.474

Sex (female)

2.02 (0.74-5.51)

0.170

0.38 (0.06-2.59)

0.326

Body mass index (kg/m²)

0.97 (0.91-1.04)

0.414

0.94 (0.83-1.06)

0.280

6-minute walking distance (m)

1.00 (0.99-1.00)

0.014

1.00 (0.99-1.00)

0.389

New York Heart Association functional class

1.56 (0.90-2.71)

0.111

3.05 (1.18-7.88)

0.021

NT-proBNP (pg/ml)

1.00 (1.00-1.00)

0.486

1.00 (1.00-1.00)

0.463

Left ventricular ejection fraction (%)

1.02 (0.98-1.06)

0.333

1.11 (1.02-1.21)

0.015

Left atrial volume index (ml/m²)

1.01 (0.99-1.03)

0.431

1.00 (0.96-1.04)

0.873

Left ventricular end-diastolic volume (ml)

1.00 (0.99-1.01)

0.531

1.01 (0.99-1.02)

0.392

E/e’ ratio

0.97 (0.89-1.04)

0.386

0.94 (0.84-1.06)

0.343

Deceleration time (ms)

1.00 (0.99-1.00)

0.185

0.99 (0.98-1.00)

0.229

Estimated glomerular filtration rate (ml/min/1.73m²)

0.99 (0.97-1.00)

0.122

1.00 (0.97-1.03)

0.991

Alanine transferase (U/l)

0.99 (0.96-1.01)

0.272

0.98 (0.93-1.03)

0.434

Aspartate transferase (U/l)

1.01 (0.97-1.04)

0.769

1.05 (0.98-1.13)

0.161

Albumin (g/dl)

0.48 (0.13-1.76)

0.269

2.26 (0.22-23.30)

0.493

Creatine kinase (U/l)

1.00 (0.99-1.00)

0.637

1.00 (0.99-1.00)

0.299

C-reactive protein (mg/dl)

1.09 (0.72-1.67)

0.674

1.35 (0.70-2.61)

0.373

Hemoglobin (g/dl)

0.90 (0.71-1.14)

0.381

0.73 (0.45-1.20)

0.212

Arterial hypertension

0.68 (0.30-1.54)

0.354

1.42 (0.37-5.47)

0.609

Diabetes mellitus

0.86 (0.41-1.79)

0.685

0.51 (0.14-1.77)

0.286

Hyperlipidemia

1.62 (0.75-3.48)

0.217

1.40 (0.32-6.15)

0.657

Smoking (ever)

2.25 (1.11-4.55)

0.024

5.59 (1.62-19.22)

0.006

Coronary artery disease

0.85 (0.42-1.72)

0.641

0.16 (0.02-1.30)

0.086

History of myocardial infarction

1.10 (0.57-2.15)

0.772

3.07 (0.57-16.49)

0.191

Atrial fibrillation

1.07 (0.49-2.37)

0.863

1.19 (0.30-4.70)

0.799

Angiotensin-converting enzyme inhibitor

1.18 (0.60-2.32)

0.628

5.13 (0.76-34.54)

0.093

Angiotensin-1 receptor antagonist

1.21 (0.60-2.43)

0.598

6.97 (0.85-57.26)

0.071

Beta-blocker

0.76 (0.26-2.23)

0.620

0.41 (0.09-1.82)

0.242

Mineralocorticoid receptor antagonist

0.89 (0.45-1.77)

0.734

0.90 (0.31-2.61)

0.847

Statins

1.30 (0.63-2.69)

0.484

1.08 (0.15-7.93)

0.938

Ezetimibe

2.56 (0.59-11.16)

0.210

6.28 (0.74-53.54)

0.093

Loop diuretic

0.66 (0.34-1.29)

0.228

0.77 (0.24-2.45)

0.655

Thiazide

0.93 (0.37-2.30)

0.868

0.69 (0.16-2.90)

0.614

Reviewer 3 Report

Comments and Suggestions for Authors

This study reports of the examination and evaluation of protein calcium-binding protein S100B in patients with chronic heart failure (HF), and the potential association of S100b changes in patients with both cardiac and cognitive dysfunctions. S100B is typically found in the brain and is a marker for blood-brain barrier (BBB) disruption.  Looking into a cohort of 146 HF patients, it was found there was a higher serum levels of S100B that linked to cardiac and cerebral sufferings with longer duration of HF, increased left atrial volume, higher prevalence of mild cognitive impairment (MCI), and lower memory scores. In spite such associations, S100B did not predict mortality, likely due to its reflection of acute injury rather than long-term outcomes. The authors conclude that S100B has potential for being a biomarker that could reflect both myocardial stress and blood-brain barrier disruption, even though the documentation that could justify this strategy obviously has limitations including its cross-sectional design, small sample size, and relatively weak associations, which may affect its clinical applicability.

The manuscript is interesting and could potentially provide novel usage of S100b also for biomarking cardiac disease. The quality of this approach would however be documented if the study included other markers specific for heart and brain degeneration. It is a bit of a wander why this was not already attempted, not at least because the authors are so convinced their study has limitations. The authors do bring this matter into discussion, but they should comment why not more markers were included and part of the correlations.  

The authors discuss that in spite of significant correlations between S100B levels and indicators of both cardiac and cognitive dysfunction, S100B might not be sufficient on its own to predict long-term outcomes like mortality. Would it be possible for the authors to suggest other markers that could have relevance in longer studies? The pathologies raised here could generate conditions with chronic inflammation and possible elevated or qualitative changes in other markers in blood, e.g. exosomes. Would their analyses be of value?

Author Response

Reviewer #3:

This study reports of the examination and evaluation of protein calcium-binding protein S100B in patients with chronic heart failure (HF), and the potential association of S100b changes in patients with both cardiac and cognitive dysfunctions. S100B is typically found in the brain and is a marker for blood-brain barrier (BBB) disruption.  Looking into a cohort of 146 HF patients, it was found there was a higher serum levels of S100B that linked to cardiac and cerebral sufferings with longer duration of HF, increased left atrial volume, higher prevalence of mild cognitive impairment (MCI), and lower memory scores. In spite such associations, S100B did not predict mortality, likely due to its reflection of acute injury rather than long-term outcomes. The authors conclude that S100B has potential for being a biomarker that could reflect both myocardial stress and blood-brain barrier disruption, even though the documentation that could justify this strategy obviously has limitations including its cross-sectional design, small sample size, and relatively weak associations, which may affect its clinical applicability.

             Thank you for this accurate summary of our findings.

The manuscript is interesting and could potentially provide novel usage of S100b also for biomarking cardiac disease. The quality of this approach would however be documented if the study included other markers specific for heart and brain degeneration. It is a bit of a wander why this was not already attempted, not at least because the authors are so convinced their study has limitations. The authors do bring this matter into discussion, but they should comment why not more markers were included and part of the correlations.

Thank you for recognizing the potential of using S100B as a biomarker for cardiac disease. The decision to focus primarily on S100B was driven by its well-established role as a marker for blood-brain barrier disruption and its emerging relevance in cardiac conditions, particularly in chronic heart failure. While we acknowledge the importance of including other markers specific to heart and brain degeneration, our study aimed to explore the novel role of S100B in this specific context. The choice not to include neurodegenerative markers was partially due to resource constraints and the desire to maintain a focused exploration of S100B. Future studies should certainly aim to include a broader panel of biomarkers to validate and expand upon our findings. We have added these thoughts to the discussion:

“Future studies should aim to include a broader panel of biomarkers to validate and expand upon our findings (…).”

The authors discuss that in spite of significant correlations between S100B levels and indicators of both cardiac and cognitive dysfunction, S100B might not be sufficient on its own to predict long-term outcomes like mortality. Would it be possible for the authors to suggest other markers that could have relevance in longer studies? The pathologies raised here could generate conditions with chronic inflammation and possible elevated or qualitative changes in other markers in blood, e.g. exosomes. Would their analyses be of value?

We agree that while S100B shows significant correlations with both cardiac and cognitive dysfunction, its utility in predicting long-term outcomes like mortality may be limited. To address this limitation, we suggest that future studies should consider including other biomarkers that could offer insights into chronic inflammation and ongoing neurodegeneration, which are relevant to the pathologies observed in chronic heart failure. Markers such as high-sensitivity C-reactive protein (hs-CRP) for systemic inflammation, exosomal proteins that may reflect ongoing neuroinflammation or cardiac injury, and brain-derived neurotrophic factor (BDNF) for neuronal health could provide valuable information in longer-term studies. These markers could potentially complement S100B and offer a more robust predictive model for patient outcomes, especially concerning long-term survival and disease progression.

By incorporating a combination of these markers, future research could better delineate the chronic processes involved in heart failure and its impact on cognitive function, ultimately improving the clinical applicability of biomarker strategies in this patient population. We have added these thoughts to the discussion:

“(…) to offer more robust predictive models for patient outcomes, especially concerning long-term survival and disease progression.”

Round 2

Reviewer 1 Report

Comments and Suggestions for Authors

The manuscript has been improved and no further recommendation. 

Reviewer 2 Report

Comments and Suggestions for Authors

Dear Editor,

authors well addressed previous comments. The paper improved very much.

Reviewer 3 Report

Comments and Suggestions for Authors

Accept